# The Network Systems Underlying Emotions: The Rational Foundation of Deep Brain Stimulation Psychosurgery

**DOI:** 10.3390/brainsci13060943

**Published:** 2023-06-12

**Authors:** Lorena Vega-Zelaya, Jesús Pastor

**Affiliations:** Clinical Neurophysiology, Instituto de Investigación Biomédica Hospital, Universitario de La Princesa, C/Diego de León 62, 28006 Madrid, Spain; lorenacarolina.vega@salud.madrid.org

**Keywords:** basal ganglia, deep brain stimulation, erethism, functional neurosurgery, limbic system, psychiatric disorders, obsessive compulsive disorder, refractory depression, thalamocortical loop

## Abstract

Science and philosophy have tried to understand the origin of emotions for centuries. However, only in the last 150 years have we started to try to understand them in a neuroscientific scope. Emotions include physiological changes involving different systems, such as the endocrine or the musculoskeletal, but they also cause a conscious experience of those changes that are embedded in memory. In addition to the cortico-striato-thalamo-cortical circuit, which is the most important of the basal ganglia, the limbic system and prefrontal circuit are primarily involved in the process of emotion perceptions, thoughts, and memories. The purpose of this review is to describe the anatomy and physiology of the different brain structures involved in circuits that underlie emotions and behaviour, underlying the symptoms of certain psychiatric pathologies. These circuits are targeted during deep brain stimulation (DBS) and knowledge of them is mandatory to understand the clinical-physiological implications for the treatment. We summarize the main outcomes of DBS treatment in several psychiatric illness such as obsessive compulsive disorder, refractory depression, erethism and other conditions, aiming to understand the rationale for selecting these neural systems as targets for DBS.

## 1. Introduction

From the point of view of neuroscience, emotions are defined as higher brain processes expressed through behavioural and somatic responses [1,2]. In this scenario, behavioural reward conditions reinforce positive reactions of pleasure, euphoria, and well-being. In contrast, reactions for avoiding punishment provoke (among other feelings) fear, sadness, pain, anger, or anxiety.

The definition of emotion can sometimes be confusing, depending on the scope of how it is analysed. There are two aspects that we must consider when we ponder emotions. First, there are automatic physiological responses, which involve changes in both cognitive and endocrine functions, as well as in autonomic and musculoskeletal functions. Second, there is the conscious experience of these changes, which are defined as feelings. The first aspect is typically used in the field of neuroscience, and the last aspect tends to be more frequently used in neuropsychology.

In this review, we summarize the main neural networks and anatomical structures implicated in affective disorders and the rationale for targeting them for deep brain stimulation surgery (DBS).

### 1.1. Understanding Emotions throughout History

Aristotle (384 B.C.–322 B.C.) stated that the centre of emotions and intelligence was the heart (Roxo MR, 2011). Centuries later, Galen (130–200), with a broader knowledge of human neuroanatomy, formulated an opposing theory. The “theory of emotions of the cerebral ventricles” postulated that the information arising from the five senses (touch, taste, smell, hearing, and sight) was processed in the cerebral ventricle by the so-called “common sense”. This process would result in a unique perception that would later travel through the following “internal senses”: fantasy, memory, and imagination. This scenario would thus generate the emotional process. For Galen, the centre of intelligence was the brain [3].

The pioneering work of Vesalius (1514–1564), known as *De humani corporis fabrica libri septem*, marked a significant leap forward in terms of neuroscientific knowledge. This work was the most complete and detailed work in this field and corrected the inaccuracies of Galen’s works. By the end of the 18th century, neurology had evolved into a more scientifically based discipline with fewer philosophical and religious aspects [4].

In the nineteenth century, with the achievements of Ramón y Cajal in identifying the neuron as the anatomical and functional unit of the central nervous system (CNS) and other advances in the fields of microscopy and cell pathology, neuroscience was established as a discipline and consolidated as a field of investigation of the complex functions and dysfunctions of the CNS. Paradoxically, the first postulate that based research on the premise that specific neurological functions could be related to a brain location was a theory in which studies did not demonstrate a concrete scientific basis. Franz Joseph Gall’s (1758–1828) “phrenology” was based on the palpation of the grooves and the shape of the skull to identify specific, physically defined functions [5]. This theory was later modified, and localizationisms were postulated with scientific rigor, first by Paul Broca (1824–1880) followed by Carl Wernicke (1848–1905) and many others.

However, the most substantial progress in understanding the cortical origin of alterations in behaviour, which was previously associated with the “soul” (i.e., the essence of human beings), was produced by the unfortunate accident in 1848 of a 25-year-old railway worker named Phineas Gage. In this case, a metal rod entered his cheek and exited through the frontal region, which shattered his skull and damaged the prefrontal cortex [6,7]. The patient, who was in perfect physical condition, developed a behavioural disorder after the accident in which he did not have the ability to use anticipatory planning and had become socially awkward [6,7,8,9,10]. Based on this event, a revolution of studies and information about the process of emotions ensued.

Meanwhile, other developments occurred. Darwin (1809–1882), with his book entitled “Expression of emotions in man and animals”, together with the pioneering American psychologist William James (1842–1910), broadened the field of affective neuroscience [11,12]. James’ theory proposed that human emotions occurred in response to afferent feedback loops from sensory receptors such as the skin, muscles, and other organs that produce physical changes that are subsequently encoded in memory in the cerebral cortex. This scenario allowed researchers to determine the subjective quality of the experienced stimuli, such as temperature change, pain, and vibration [13].

In 1894, Sigmund Exner (1846–1926) described a neural circuit model based on animal experiments, which explained the interactions between feelings of pleasure and aversion in the brain. Exner postulated that sensory events acquired emotional significance and produced motor and autonomic responses. This model anticipated current neurobiological theories [14] and postulated that the thalamus would function as a sensory integration centre and as a filter that would direct only intense stimuli to a specific structure comprised of neuronal cell bodies. Later, this aversive centre that Exner was referring to would become known as the amygdala.

By the middle of the twentieth century, experiments began to differentiate the psychological and neurophysiological domains from emotional processes [15]. In 1920, investigations with longitudinal sections of animal diencephalons conducted by Walter Cannon and Philip Bard defined the emotional brain as being the caudal half of the hypothalamus and the posteroventral thalamus [1,5].

In 1939, Kluver and Bucy demonstrated that the bilateral removal of the temporal lobe in monkeys (including the amygdala and hippocampal formation, as well as the nonlimbic temporal cortex) produced a particular behavioural syndrome [16]. Specifically, the monkeys became docile and exhibited a reduced emotional threshold. Moreover, they demonstrated hypersexualized behaviour and a tendency towards oral behaviour, such as attempting to ingest inedible objects [17,18]. Today, this syndrome is known as Kluver–Bucy syndrome and is the first evidence that the limbic system is responsible for the cortical processing of emotions. In 1937, the American neuroanatomist Papez described a circuit of brain connections that generates emotion as a result of the flow of information travelling through reciprocal anatomical networks between the hypothalamus and the mesial cerebral cortex. This circuit is known as the Circuit of Papez [19]. According to Papez, there are two pathways for the integration of neural information: one occurs through the hippocampus and the cingulate cortex, which is directly involved with hypothalamic activity, and the other pathway occurs through the lateral cortex, which is involved in nonspecific sensory activities mediated by the dorsal thalamus. This circuit also includes other brain structures with locomotor, memory, and associative functions [19]. Through this flow of nervous stimuli, sensations are transformed into perceptions, thoughts, and memories. In addition, Papez believed that the participation of the cerebral cortex in the subjective processing of emotions was essential.

From these observations and until now, the functioning of emotions has been considered a complex system in which multiple anatomical structures and their connections are involved.

Acronyms and full term descriptions are included in the Abbreviations section.

### 1.2. Neuroanatomical Structures and Networks Involved in Emotions

In the same manner that movement disorder surgery was developed from clinical observations, psychosurgery has evolved from information and research on patients with changes in cognitive and affective behaviour after brain injuries. In fact, the application of DBS in psychiatric diseases constitutes the greatest progress in functional neurosurgery. Therefore, a proper understanding of the current targets used in psychiatric disorders and a basic anatomical knowledge of cortical and subcortical structures and the circuits that are involved are mandatory for DBS.

### 1.3. Basal Ganglia: The Main Modulatory Structures

The basal ganglia (BG) refers to the putamen, caudate nucleus, globus pallidus, subthalamic nucleus, and substantia nigra. However, to simplify the description of its connections with other brain structures, other names have been designated that represent different combinations of BG. These arrangements have been determined either by their common embryonic origin (caudate/putamen = striate) or by their common connections (putamen/globus pallidus = lenticular nucleus).

The caudate and putamen maintain embryological continuity just above the orbital surface of the frontal lobe, wherein the caudate head appears to be continuous with the anterior part of the putamen. This region of continuity is known as the nucleus accumbens (NA), which, together with adjacent parts of the caudate and putamen, forms a separate division known as the ventral striatum (VS).

The cortico-striate-thalamo-cortical circuit (CSTC), which is the most important of the BG, is the main network involved in movement; therefore, its malfunction is responsible for several movement disorders, such as dystonia or tremor. In addition, they can also cause behavioural and emotional disorders. The cortico-thalamic (CT) loops begin their projections from an area of the cerebral cortex (usually in the frontal lobe or limbic cortex) to a specific target area within the thalamus. Parallel to this direct CT pathway, there is a topographically organized circuit that involves the same regions of the cortex and the thalamus but is directed through the BG. Specific cortical areas project to the striatum (which is the input nucleus of the BG). The output fibres from the striatum to the globus pallidus are inhibitory, along with those fibres originating from the substantia nigra reticularis (SNr). Efferent pathways to the thalamus remove thalamic inhibition, thus producing excitatory connections to the cortex (Figure 1).

Recently, it has been possible to delimit the flow of information through these networks and the interplay between the thalamus and the cortex involved in the appearance of certain behaviours [20].

The BG can act as a filter and control the expression of learned automatic behaviours versus responses to a novel stimulus. In this manner, these structures provide an anatomical substrate to surgically modulate brain function, thus making CT/CSTC loops the primary targets when considering DBS for psychiatric illness.

### 1.4. Limbic System and Hypothalamus

In general, we rarely perceive things in a completely neutral way. Certain sights or sounds make us feel happy, sad, or angry. There is also a bi-directional connection between both these emotional perceptions and thoughts and memories. Some scents can remind us of images of food or drinks. Moreover, memories of a particular event from the past can elicit the same emotion that accompanied the original event. It is known that the action of having a thought depends extraordinarily on the neocortex; thus, it is logical to assume that any anatomical substrate that deals with feelings and emotions must have a close relationship with the neocortex. However, in addition, it would also need to be closely related to the hypothalamus, due to the fact that the sensory stimuli that arouse an emotion also initiate an autonomic response, including salivation, increased adrenaline, and shunting of blood to skeletal muscles, among other responses. The type of stimulus that causes an emotion with a particular response is critical for all animals because it involves fundamental behaviours for the preservation of individuals and their species, such as feeding, defence, and sexual behaviour.

The brain regions that are primarily involved with the aforementioned responses and behaviours form the limbic system. In general, this region is considered to be composed of the cingulate cortex, the parahippocampal gyrus, the amygdala, and the hippocampus. However, there are other structures also involved in this process. For this reason, the limbic system is more of a functional system than an anatomically restricted system. In fact, the connections between these anatomical structures compose what is known as the limbic circuit. The limbic loop is a network consisting primarily of the medial orbitofrontal cortex (MOFC), anterior cingulate cortex, agranular insular cortex, and their associated projections through the anterior internal capsule (AIC) and substantia innominate to the caudate and NA [21] (Figure 2).

There are direct and reciprocal glutamatergic connections between these regions and the posterior and medial portion of the dorsomedial (DM) thalamic nucleus via fibres travelling in the AIC and extending posteriorly to the inferior thalamic peduncle [22]. Furthermore, it has been described that posterior parts of the DM nucleus may be inherently associated with the centrolateral intralaminar (CL) nucleus [23].

In parallel with the direct connections to the thalamus, another connection passes through the BG. The input to the striatum from these cortical regions projects almost exclusively to the ventromedial striatum (VMS), which consists primarily of the more ventromedial portions of the caudate, as well as the NA. From the VMS, a series of projections are directed to other BGs.

The NA is divided into two functionally distinct components: the shell and the core. The NA core (NAc) is the dorsolateral component and is distinguished by a lower concentration of µ-opioid receptors and a higher concentration of calcium-binding proteins. In contrast, the NA shell (NAs), which is ventromedial to the NA, lacks calcium-binding proteins and is replete with µ-opioid receptors. The projections of the cortical regions are segregated according to the division of the NA. The central component of the ventromedial caudate/NAc receives projections from all of the previously identified cortical regions. However, the NAs receive a set of projections from the subgenual cingulate and the pregenual cingulate. These striatal regions correspondingly keep their efferents segregated. Thus, efferents from the NAc project to the ventromedial GPi and dorsal SNr/ventral tegmental area (VTA). The NAs project in a similar fashion to the dorsal SNr/VTA but maintain a separate branch to the ventral pallidum. Ultimately, all of these structures mainly project towards the DM magnocellular thalamus to close the loop.

Another structure that is significantly associated with the limbic system is the amygdala. It is strongly involved in the production of emotional states and behaviours. In addition, it plays a particularly important role in the regulation of conditioned fear and anxiety responses [24,25]. The amygdala connections are extensive and parallel the striatal component of the CSTC limbic loop (Figure 3).

The main efferent pathways from the amygdala are: (i) the stria terminalis towards the septal area(s) and the hypothalamus; (ii) direct projections to the hypothalamus, thalamus, and extensive areas of the frontal and insular cortex and olfactory structures; and (iii) direct projections to the hippocampus, entorhinal, and temporal cortex. Afferent pathways to the amygdala originate from four regions: (i) from the hypothalamus and septal nucleus(s) via the stria terminalis; (ii) from the thalamus and hypothalamus, as well as from the orbital and cingulate cortex; (iii) from the cortex and olfactory bulb; and iv) directly from the temporal lobe.

### 1.5. Dorsolateral Prefrontal Cortex (DLPFC) and Lateral Orbitofrontal Cortex (LOFC)

The DLPFC and LOPC form part of the prefrontal circuit. Their projections through the AIC and the inferior thalamic peduncle (ITP) are directed towards the thalamic nuclei ventral anterior parvocellularis and magnocellularis (VApc and mc) and the lateral portion of the dorsomedial (DM). There are also connections to the BG, particularly to the dorsolateral striatum, which is considered the motor striatum, as well as to the central striatum (CS) or associative striatum and to the ventromedial region (VMS) or limbic striatum. The input regions of this circuit involve the caudate head and the central and rostral portions of the putamen [26]. These striatal regions correspondingly project to the outlet structures of the dorsomedial GPi and the rostrolateral SNr. The BG output structures mainly project to the specific thalamic relay nuclei VApc and parvocellular portion of the dorsomedial thalamic nucleus, as well as to the intralaminar nucleus (Figure 4).

The prefrontal circuit is part of the processes that involve working memory, spatial memory, and executive function. Additionally, it plays a role in cognitive dysfunction, the ability to suppress negative feelings and painful stimuli, which are observed in some psychiatric disorders. In addition, the psychomotor retardation of severe depression may also result from the dysfunction of this circuit [26,27,28,29].

Another cortical structure that deserves special mention is the orbital frontal cortex (OFC), which receives projections from the amygdala, anterior cingulate, DLPFC, hippocampus, insula, posterior hypothalamus, and periaqueductal grey matter. Its simultaneous relationship with the striatum, which allows it to influence motor behaviour in addition to its regulatory role in the autonomic nervous system, makes it a central integrator between the internal state (hypothalamus) and the external state (striatum) [30].

### 1.6. Subthalamic Area

In addition to the striatum, the subthalamic nucleus (STN) is an important input to the BG. Not only is it a primary nucleus for movement function, but it also has an area associated with limbic and associative circuitry. The STN itself and its adjacent medial subthalamic region (MSR) extend to the wall of the third ventricle to the subthalamic area. Three areas in the MSR have been described as targets in different psychiatric pathologies: the anteromedial STN, the medial forebrain bundle (MFB) and the “Sano triangle” [31,32,33,34,35,36]. These regions have special neuronal characteristics and form a transition zone between the basal ganglia and the limbic system [37].

The STN is the structure that controls both the entry to the striatum through its dopaminergic connections with the substantia nigra compacta (SNc) and the exit of the BG through its connections with the SN reticular (SNr) and the GPi (Figure 1). Therefore, it plays a major role in synchronizing the oscillatory activity of the BG and the cerebral cortex [38,39].

## 2. Clinical Correlations and DBS Targeting

Neuromodulation of specific brain networks in psychiatric illness is supported by findings that suggest that these conditions are (at least in part) caused by disturbances in the previously mentioned circuitry.

Although it has experienced growing interest, DBS in psychosurgery still does not have conclusive evidence of its therapeutic efficacy. Perhaps the most important reason for this scenario is that, from an anatomical point of view, the therapeutic targets are not clearly defined. Therefore, its functional delimitation would likely be the most useful tool to improve clinical results [40,41,42].

### 2.1. Obsessive Compulsive Disorders (OCD)

OCD affects 2–3% of the population [43]. Approximately 10–30% of patients do not respond to combined treatment with cognitive-behavioural and pharmacological psychotherapy, including serotonin reuptake inhibitors, clomipramine, and atypical antipsychotics.

This disorder is a very complex disease that is characterized by irrepressible behaviours with intense impulsiveness, compulsions, and rituals; however, it also presents with intense motor inhibition, due to the fact that the obsessions lead to a paralyzing doubt [44].

A circuit-based model of OCD suggests that obsessive compulsive symptoms appear when striato-pallido-thalamic activity is abnormally decreased or when orbitofrontal-thalamic activity is abnormally increased [45]. Accordingly, OCD symptoms can occur when striatal-pallido-thalamic activity decreases and fails to modulate activity in the positive fronto-thalamic loop [26].

Additionally, functional and structural neuroimaging studies have established that the symptomatology of OCD is particularly mediated by abnormalities in two relatively segregated frontostriatal loops: the affective loop and the space/attention loop. The affective loop includes the orbitofrontal cortex, the NA, the ventral pallidum and the thalamus, with alleged influences of the anterior cingulated cortex, the hippocampus, and the basolateral amygdala. Dysregulation of the affective loop in OCD is related to deficits in the representation of reward, punishment, anxiety, emotional processing, and inhibitory control. In contrast, the space/attentional loop includes the dorsolateral prefrontal cortex (DLPFC), the caudate nucleus, the pallidum, the thalamus, and the SN. Alterations in this circuit seem to be related to deficits in executive planning, cognitive flexibility, implicit learning, and working memory [46].

Neurosurgical treatment in refractory cases has been performed for a long period of time by using empirically defined targets, such as the AIC. Anterior capsulotomy can benefit 35–70% of these patients; however, the irreversibility of the lesions often discourages the use of this procedure. The first data on BDS in AIC, published by Nuttin et al. in 1999 [47], showed a response rate of 50%. The conducted studies suggested that stimulation of the AIC works in OCD by influencing the activity of the near limbic ventral striatum. Thus, DBS to the AIC was approved by the FDA (Humanitarian Device Exemption) in 2009 [48].

In more recent years, STN stimulation has demonstrated favourable results, with an improvement in overall postoperative functioning [49]. The anteromedial STN primarily receives input from the ventral limbic pallidum [50]. It also receives multiple sources of innervation, with little dopaminergic projection from the SNc and ventral tegmental area [51,52,53] and significant serotonergic and cholinergic innervation. If modulatory inputs exist and are restricted to specific territories in the nucleus, an understanding of the clinical effects observed after DBS would yield important insights into the physiological basis of OCD.

However, other targets have been proposed, including the NA [54], ventral capsule/ventral striatum [55], inferior thalamic peduncle [56], and medial dorsal and ventral anterior nuclei of the thalamus [57]. In other recent prospective clinical studies, electrodes were implanted in several targets per patient using the STN, anterior internal capsule, NA, and ventral striatum [58,59].

### 2.2. Refractory Depression

Depression is one of the most frequent psychiatric disorders, with an estimated prevalence of 3–17%. It is one of the fundamental causes of global disability, is highly recurrent, and is associated with a high mortality rate [60]. Moreover, it is estimated that 15–30% of patients with major depression do not respond to conventional therapies.

Much of the research implicating the basal ganglia and other structures in the pathogenesis of affective disorders originates from analyses of imaging tests, such as positron emission tomography (PET) and functional magnetic resonance imaging (fMRI). However, it is important to mention that neuroimaging is hampered by the heterogeneity of affective disorders; nevertheless, some features have shown some consistency.

Depression symptoms are associated with abnormal activity in the ventral striatum, as well as in the orbitofrontal and subgenual cingulate cortices [61]. It seems that there is increased metabolism in the orbitofrontal, anterior insular, and subgenual cingulate cortices. Other findings indicate a decreased metabolism in the DLPFC. Finally, in depressed patients, elevated metabolism in the amygdala has been observed, which has been reversed with the use of pharmacotherapy [62].

Previous studies have suggested that a dysfunction in connections from the cingulate cortex to the dorsal (including the DLPFC, inferior parietal cortex, and striatum) and ventral parts (hypothalamic–pituitary–adrenal axis, insula, subgenual cingulate, and brainstem) is associated with an alteration in the emotion regulation circuit and consequently involved in depression [63]. Long-term data on DBS for depression have been reported in targeting the subgenual cingulate cortex [61,64,65,66].

Another structure that has been used for DBS in depression is the NA [67,68]. Specifically, the NAc has been observed as the link between the systems involved in emotion and motor control. In addition, anhedonia is correlated with NAc dysfunction [69].

The supero-lateral branch of the MFB has also been proposed as a target [70]. The MFB is a bundle of fibres from the MSR that contains both ascending and descending fibres from the mesolimbic pathways. These fibres are mainly dopaminergic and serotonergic [71]. The mechanism that explains its clinical benefit could be that high-frequency stimulation activates the basal structures of the forebrain and prefrontal cortex [37]. It is possible that the activation of this structure from its dopaminergic fibres could mediate both alleviation of mood and anti-OCD effects.

Other targets that are used with a good clinical response are the ventral striatum and the AIC [72].

### 2.3. Aggressiveness

Pathological aggressiveness is an important social problem, due to the fact that it is estimated that it affects up to 45% of patients with mental disabilities [73].

DBS of the posteromedial hypothalamus (PMH) has been proposed as a treatment for resistant erethism, although experience with this treatment throughout the world is scarce; in addition, there are studies with case series that have demonstrated very good results [74,75,76,77]. Sano’s triangle (as a target for aggressiveness in schizophrenic and autism-spectrum disorder patients with stereotactic lesions) was described by Sano between 1960 and 1970 and included the PMH [34,35,78]. From an anatomical point of view, the triangle of Sano lies posterior to the mammillary body, which is actually defined as the posterior limit of the hypothalamus [79,80]. It does not have clearly defined boundaries but lies between the posterior hypothalamus and the periaqueductal grey midbrain [81,82].

In the literature, high levels of oestrogen receptor α type (ERα) expression have been described in the hypothalamus and in other areas of the limbic brain, such as the amygdala, basal forebrain, and mammillary body [83,84]. Experimental animal studies have demonstrated the role of hypothalamic ERα neurons in aggressive behaviour, with the lack of the ERα gene severely reducing the production of aggressive behaviour in male mice [85]. The use of optogenetic techniques within the ventromedial hypothalamus of mice has shown that the activation of ERα+ neurons could immediately generate aggression. Interestingly, weaker optogenetic activation of these specific neurons favours sexual behaviour [86].

Recently, the presence of specific atypical action potentials in the PMH has been described and can help to more precisely define the target [41,42] during microelectrode recording.

### 2.4. Anorexia Nervosa (AN)

There are few effective and long-lasting treatments available for refractory anorexia nervosa (AN); unfortunately, conventional treatment fails to achieve a response in approximately 50% of patients.

In recent decades, studies have been performed that provide a better understanding of the neural circuitry of AN. Structural and functional imaging tests indicate a highly heterogeneous pathology with a high rate of comorbidity with other affective disorders. For example, alterations in the subcallosal cingulate cortex, which has been implicated in the network of other affective disorders [87], have been shown to alter serotonin binding in patients with acute anorexia [88]. Structures that are crucial to reward and perception pathways, such as the ventral striatum and temporoparietal junction, have also been implicated in imaging studies, thus suggesting dysfunction through multiple overlapping circuits. Current conceptions of AN circuitry postulate the existence of key modulatory centres, including the insula and cingulum subcallosum, which modulate effects on cognition and overt behaviour [89]. These models provide the rationale for functional neurosurgical treatment in patients with refractory AN.

### 2.5. Drug Addiction

The initial studies that led to the investigation of DBS for addiction in humans were motivated by observations of Parkinson disease (PD) patients who were treated with DBS. Several case series have reported that DBS in the subthalamic nucleus could curb symptoms of dopamine dysregulation syndrome (DDS) [21,90], which is a condition characterized by neuropsychiatric disorders such as psychosis, pathological gambling, hypersexuality, and mood changes.

In the same sense, DBS treatment in the NAc in patients with diagnoses of anxiety, depression and OCD has resulted in cases of at least improvement or (in many others) of the complete disappearance of drug dependence in those patients who exhibited this condition as an added comorbidity [91,92]. The NAc represents the main target of interest in addiction not related to PD, but also for other psychiatric illnesses with an addiction disorder as a comorbidity.

## 3. Discussion

DBS is an effective and proven surgical treatment for several movement disorders and seems to be promising for psychiatric pathologies [93]. The process for all of these diseases is to implant electrodes at different targets to modify the pathological behaviour of neural circuits by means of electrical stimulation. Therefore, a better knowledge of the anatomical interconnectivity of pathophysiologically relevant cortical and subcortical areas will refine and increase the applications and effectiveness of DBS for the treatment of psychiatric pathologies.

In recent decades, imaging techniques (i.e., computer tomography or magnetic resonance imaging) have experienced an impressive expansion, and it has been argued that imaging-guided surgery can increase the targeting accuracy in STN and internal globus pallidus (GPi) surgeries, thus improving the clinical outcomes in patients with PD. However, a very important aspect in DBS in general and particularly in psychosurgery is the correct identification of the target to achieve the best therapeutic effects, thus minimizing the side effects and optimizing the battery. This aspect becomes more important if we consider that some of the target areas do not have a well-defined anatomical limit.

As we have shown in this study, the BG circuits include different parallel systems, and single-cell recordings in animals have been shown to preserve functional specificity at the level of individual neurons throughout circuits [94]; therefore, the objective should not merely be to target a nucleus as a whole but also to stimulate specific locations inside of the nucleus. Moreover, some neural structures, such as the thalamus or the hypothalamus, are highly complex, lacking anatomical landmarks that discriminate between subnuclei. Although it has been proposed that diffusion tensor imaging can be useful in identifying white pathways [64], not all of the relevant targets possess these long pathways. However, some thalamic [95,96,97] or hypothalamic [36,41,42] subnuclei exhibit bioelectrical-specific features that allow confidence in identification, even under general anaesthesia. An unequivocal method to identify the thalamic ventrocaudal subnucleus is via the MER of somatosensory evoked potentials [40]. The positive identification of this structure makes any type of cerebro-spinal fluid drainage or brain shift irrelevant and helps to identify surrounding nuclei [98].

However, some of the networks involved in emotions include cortical and white matter fibres, in addition to deep nuclei. The identification of these structures would likely be achieved more efficiently by means of neuroimaging than through the use of MER.

In any case, a deeper understanding of the neural networks implied in emotions, as well as the easiest and more efficacious identification of these targets, will improve the accuracy of DBS, thus expanding upon its use to many patients and many pathologies.

Knowledge of neural networks is also excellently useful for the implementation of other novel therapies in psychiatric pathology. In 2014, a group of European experts published a series of guidelines on the therapeutic use of repetitive transcranial magnetic stimulation (rTMS) for, among other disorders, depression, anxiety disorders, OCD, schizophrenia, anxiety/addiction, and conversion [99]. A new updated version was published in 2020. Level A evidence (definite efficacy) was reached for: high frequency (HF)-rTMS of the left DLPFC using a figure-of-8 or a H1-coil for depression; HF-rTMS of the right DLPFC in posttraumatic stress disorder; low frequency (LF)-rTMS of the right DLPFC in depression; and bihemispheric stimulation of the DLPFC combining right-sided LF-rTMS (or continuous theta burst stimulation) and left-sided HF-rTMS (or intermittent theta burst stimulation) in depression [100]. Although no studies have been found that support a level A efficacy for rTMS in patients with OCD, since 2018 the FDA has approved the use of deep rTMS as an adjunct for the treatment of adult patients with OCD. The studies with the best results stimulate the prefrontal cortex and the anterior cingulate bilaterally [101,102].

## Figures and Tables

**Figure 1 brainsci-13-00943-f001:**
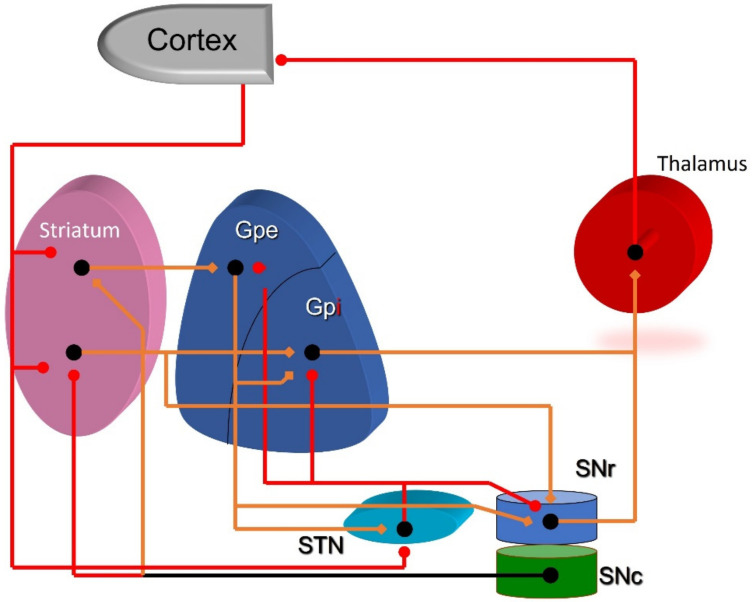
Cortico-striate-thalamo-cortical circuit. Excitatory connections: red. Inhibitory connections: orange. Black line represents the dual effect (either excitatory or inhibitory) of SNc projections. Gpe: globus pallidus pars externus; Gpi: globus pallidus pars interna; SNc: substantia nigra compacta; SNr: substantia nigra reticularis.

**Figure 2 brainsci-13-00943-f002:**
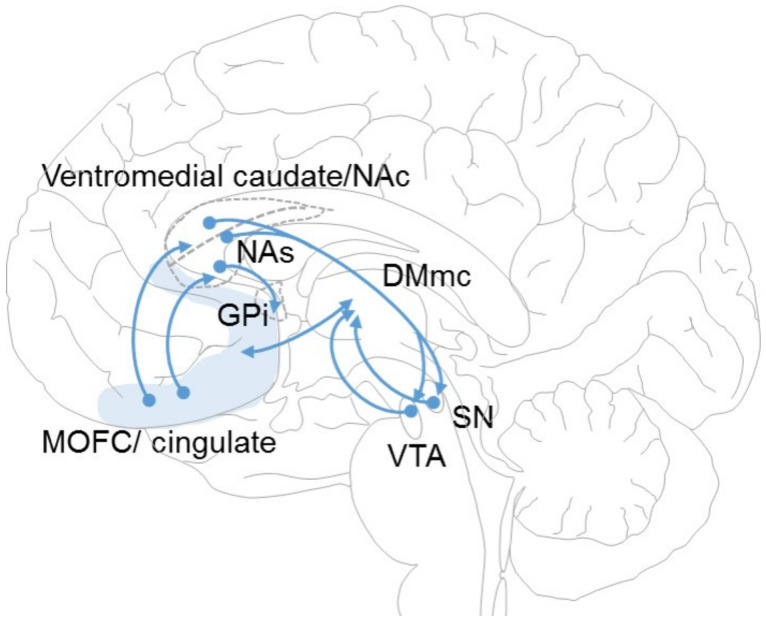
Scheme of the limbic corticostriatal-pallidothalamocortical (CSPTC) loop in a medial vision of the encephalon. ITP: inferior thalamic peduncle; DMmc: dorsomedial thalamic nucleus magnocellular portion; GPi: globus pallidus pars interna; MOFC: medial orbitofrontal cortex (shaded); NAc: nucleus accumbens core; NAs: nucleus accumbens shell; SN: substantia nigra; VTA: ventral tegmental area.

**Figure 3 brainsci-13-00943-f003:**
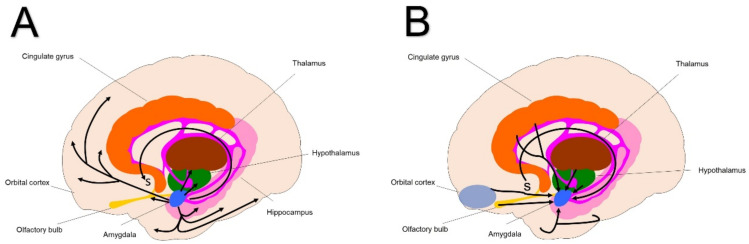
Connections of amygdala and related structures. (**A**) Efferent pathways. (**B**) Afferent pathways. See main text for details. s: septal area.

**Figure 4 brainsci-13-00943-f004:**
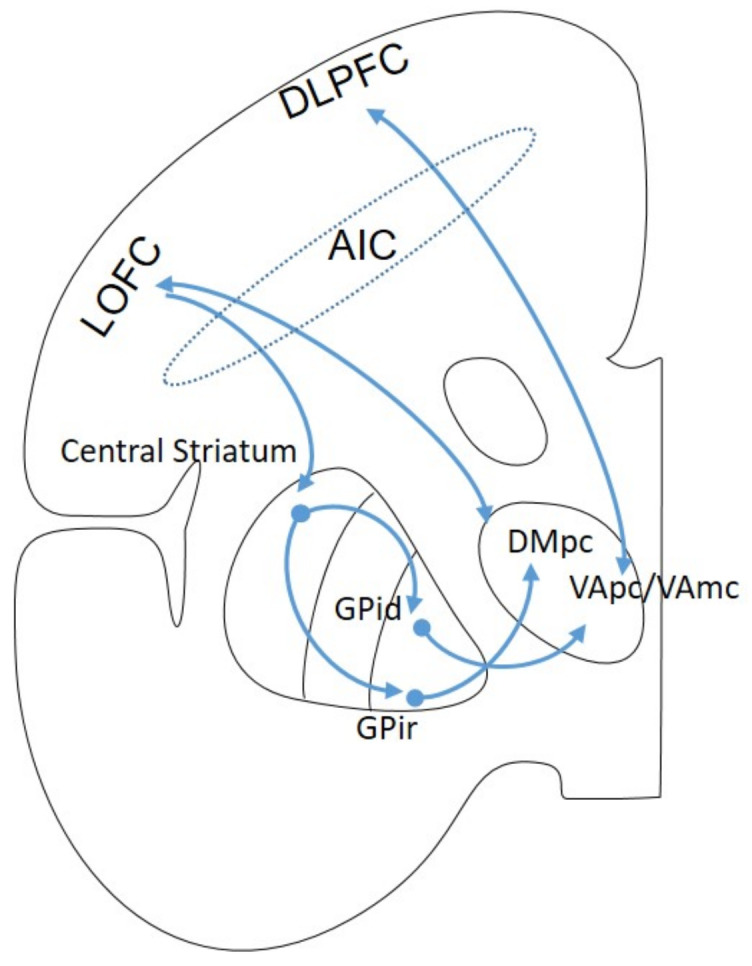
Scheme of the prefrontal loop in a frontal section of the brain. DLPFC: dorsolateral prefrontal cortex, LOFC: lateral orbitofrontal cortex, GPi: globus pallidus dorsal; GPir: globus pallidus rostral, VAmc: ventral anterior thalamic nucleus magnocellular portion, VApc: ventral anterior thalamic nucleus parvocellular portion, DMpc: dorsomedial thalamic nucleus parvocellular portion; AIC: anterior internal capsule.

## Data Availability

Data sharing not applicable.

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
