# Peer review of "The Network Systems Underlying Emotions: The Rational Foundation of Deep Brain Stimulation Psychosurgery"

_brainsci, 2023, doi:10.3390/brainsci13060943_

Round 1

Reviewer 1 Report

The aim of the paper is to present the network of underlying emotions but it still lacks a broader presentation on psychosurgery, indications, results, and side effects.  The reader is shifted to the historical perspective, and pure anatomical presentation and projections are given with too many acronyms introduced., Therefore, if the reader is not familiar with details from the current indications and the most common psychosurgical procedures that are in use today, it would be somehow hard to digest the current paper. The review should be a compact unit, therefore, covering both aspects the psychosurgery (in the title of the submitted paper) and anatomical systems underlying emotions.

-It is suggested to make a diagram relating to the projections /raw 194-232. It is hard to visualize all the anatomical details, and Fig 2 -Fig 3 are insufficient to follow the text. Also, the 1.6 paragraph does not include any visualization of the STN. Therefore from 1.4 paragraphs to 1.6 paragraphs, it would be suggested to somehow link all the areas and projections by making a diagram for the reader to follow more easily. Please include full terms and acronyms.

-Raw 15- “This circuit of emotions” should be clearly rewritten to be more understandable.

-The source for Figure1-Fiure 4 should be clearly stated.

-anterior internal capsule (AIC) is not depicted in the Fig 2 and in the Fig 2 legend it is included as full term and acronym

-Fig 3 clearly states which color is devoted to which area

-In total, the text has too many acronyms, and it is difficult to read.

-ECP stands for? PD? The full term is missing.

-Raw 419  “dot” before Discussion title – the entire Discussion needs total rewriting covering the part that is missing on the psychosurgery procedures used today and indications and results.

-The aim of the study, Introduction and the Discussion need total rewriting.

Author Response

Dear reviewer:

Thank you for the comments concerning our manuscript. Those comments are all valuable and very helpful for revising and improving our paper. We have studied comments carefully and have made correction which we hope meet with approval. We will reply one by one as follows and the part that replies to you is marked in red of all revisions in the revised manuscript.

The aim of the paper is to present the network of underlying emotions but it still lacks a broader presentation on psychosurgery, indications, results, and side effects.  The reader is shifted to the historical perspective, and pure anatomical presentation and projections are given with too many acronyms introduced. Therefore, if the reader is not familiar with details from the current indications and the most common psychosurgical procedures that are in use today, it would be somehow hard to digest the current paper. The review should be a compact unit, therefore, covering both aspects the psychosurgery (in the title of the submitted paper) and anatomical systems underlying emotions.

The objective of our article is to review the anatomical structures and the connections that underlie and produce emotions. We believe that the description and correct understanding of these systems is essential to justify the targets used in psychosurgery. The indications, results and side effects of deep brain stimulation in psychiatric patients go beyond the objective of our review, in fact we believe that reaching this new objective would deserve a new separate review article.

-It is suggested to make a diagram relating to the projections /raw 194-232. It is hard to visualize all the anatomical details, and Fig 2 -Fig 3 are insufficient to follow the text.

This is a quite interesting suggestion, but anatomical details are not searched by these figures. In fact, our goal is to schematically show the connections between structures and, we believe that a single figure composed of Fig 2 and 3 would be even harder to visualize.

Also, the 1.6 paragraph does not include any visualization of the STN.

We have referred in line 273 to Figure 1.

Therefore from 1.4 paragraphs to 1.6 paragraphs, it would be suggested to somehow link all the areas and projections by making a diagram for the reader to follow more easily.

Although your point of view is quite reasonable, we think that such a extremely complex figure results more difficult to understand.

Please include full terms and acronyms.

Thank you for your suggestion. We have included acronyms at the Appendix.

-Raw 15- “This circuit of emotions” should be clearly rewritten to be more understandable.

Done.

-The source for Figure1-Fiure 4 should be clearly stated.

Figure 1 and 3 is not taken from any source. All the diagrams are original from the authors.

You can follow the detailed description in reference [22] Kopell BH, Greenberg BD. Anatomy and physiology of the basal ganglia: implications for DBS in psychiatry. Neurosci Biobehav Rev. 2008;32(3):408-22.

-anterior internal capsule (AIC) is not depicted in the Fig 2 and in the Fig 2 legend it is included as full term and acronym

Corrected

-Fig 3 clearly states which color is devoted to which area

The different regions from where amygdala receives and project to connections are clearly indicated by arrows and colored structures/areas are named. Therefore, a description of color code would be redundant.

-In total, the text has too many acronyms, and it is difficult to read.

Your comment is right but is the only way to manage so many structures, nuclei and bundles repeatedly mentioned along the text. We have included a detailed description at Appendix.

-ECP stands for? PD? The full term is missing.

There has been an error in the acronym, it has been corrected in all lines and the PD has also been defined.

-Raw 419  “dot” before Discussion title – the entire Discussion needs total rewriting covering the part that is missing on the psychosurgery procedures used today and indications and results.

We have removed the dot. As stated at Introduction (lines 39-41), our goal is not the detailed analysis of psychosurgery procedures but “In this review, we summarize the main neural networks and anatomical structures implicated in affective disorders and the rational to target them for deep brain stimulation surgery (DBS).” Obviously, your suggestion is right and deserves a different article, but was not the aim of this.

-The aim of the study, Introduction and the Discussion need total rewriting.

If we change the objective of the article, this work will become a completely different article. This does not mean that one or the other would be incorrect from a scientific point of view, it is simply that they would be two works that yield different scientific information.

Reviewer 2 Report

This review article focused on the anatomy and physiology of various brain structures involved in emotions and behavior and how they related to several psychiatric conditions. In addition, a major goal of the review seemed to be how the basic anatomy and physiology of these areas has implications for deep brain stimulation (DBS). This is because research has been done on DBS for these areas. Thus, the authors maintained that knowledge of the anatomy and physiology of the various brain regions covered in the review (BG, Limbic System, Hypothalamus, DLPFC, LOFC) may have implications for DBS of these regions in disorders such as (OCD, depression, Erethism, AN, and drug addiction).

Overall, I think this review was relatively well-written, covers an important topic, and provided a great deal of interesting information on the above topics. I think it will be of interest to readers of the journal and adds to the literature on the topic. Thus, I think the paper should eventually be published after revisions are made. I do have one relatively major comment along with a number of minor issues that should be addressed below.

Major Comment:

Based on the purposes of the review, I strongly feel that the authors need to cover the available literature more and in much greater detail on the studies that have involved DBS of these areas. I know that there is not nearly the literature on these areas in regard to DBS as in Parkinson’s disease or dystonia. Plus, the authors do cite a few studies on these topics. However, since this is a review article I feel more literature on these topics is needed and the studies described in much greater detail. Relatedly, the review reads more like a textbook than a review article for that reason, but also due to the long history section at the beginning. Thus, I think more of the research on DBS of these areas needs to be covered and more in line of what a review article would have as opposed to a textbook type of writing style.

Minor comments:

Overall, the article was well written. However, there are some instances of various types of typos, wording, and writing style as well as formatting that should be changed.

1.      In a large number of places in the paper some words are in a different font than the others. Here are some examples that appear to be a different font, although there are likely more that I missed. Please correct these and any other instances of this.

a.       Line 30 “In contrast”

b.      Line 64 “Gall’s”

c.       Line 102 “behavior”

d.      Line 117 “has been”

e.       Line 234 “form”

2.      There are many instances in the paper where the authors have a paragraph that is one sentence. There are a few others with 2 sentences. A paragraph in general shouldn’t be one sentence. These can be combined into another paragraph and would logically fit in one of the surrounding paragraphs. Considering changing these. Examples include paragraphs associated with lines 39-41, 117-119, 152-154, 277-278, 290-292, 363-364, 366-367, 387-389, 391-393, and 453-455.

3.      A few times the manuscript is referred to as “this study”. I think referring to it as “this review” would be better. Examples are lines 39 and 435.

4.      Bibliography: spacing in line 536 is off. Furthermore, some articles have all caps for titles of studies (e.g. referencse 42, 57, 58, 59, 64) whereas most others do not. Check all bibliography formatting.

5.      Lines 68-76. Why not call him Phineus Gage by name since this is such a historic example. Students may not have heard of this and want to find out about the story. Mentioning his name would be helpful.

6.      Line 140 should “striate” read “striatal”??

7.      Figure 1. Why does the black line from the SNc change to orange? Why not just be orange the whole way. Better yet since this pathway has two branches and is excitatory to the direct pathway and inhibitory to the indirect pathway of the striatum (due to different post synaptic receptors) why not have 2 separate lines the whole way from the SNc??? In addition, wouldn’t it be better to point out the direct, indirect, and hyperdirect pathways on this figure? They are depicted but not labeled. Based on the importance of knowing the physiology that this article discusses it seems like that should be on there.

8.      Figure 3. Would “afferents” or “afferent pathways” be better to put on the figure than “afferences”? Analogous comment for efferences.

9.      If possible figures 2, 3, and 4 could be bigger.

this was covered in my review

Author Response

Dear reviewer:

Thank you for the comments concerning our manuscript. Those comments are all valuable and very helpful for revising and improving our paper. We have studied comments carefully and have made correction which we hope meet with approval. We will reply one by one as follows and the part that replies to you is marked in red of all revisions in the revised manuscript.

This review article focused on the anatomy and physiology of various brain structures involved in emotions and behavior and how they related to several psychiatric conditions. In addition, a major goal of the review seemed to be how the basic anatomy and physiology of these areas has implications for deep brain stimulation (DBS). This is because research has been done on DBS for these areas. Thus, the authors maintained that knowledge of the anatomy and physiology of the various brain regions covered in the review (BG, Limbic System, Hypothalamus, DLPFC, LOFC) may have implications for DBS of these regions in disorders such as (OCD, depression, Erethism, AN, and drug addiction).

Overall, I think this review was relatively well-written, covers an important topic, and provided a great deal of interesting information on the above topics. I think it will be of interest to readers of the journal and adds to the literature on the topic. Thus, I think the paper should eventually be published after revisions are made. I do have one relatively major comment along with a number of minor issues that should be addressed below.

Major Comment:

Based on the purposes of the review, I strongly feel that the authors need to cover the available literature more and in much greater detail on the studies that have involved DBS of these areas. I know that there is not nearly the literature on these areas in regard to DBS as in Parkinson’s disease or dystonia. Plus, the authors do cite a few studies on these topics. However, since this is a review article I feel more literature on these topics is needed and the studies described in much greater detail.

Thank you for your comment and suggestion. However, the movement disorders are clearly out of the scope of our review, focused on neural networks of emotions. Therefore, Parkinson’s disease and dystonia are not included in our research interest.

Relatedly, the review reads more like a textbook than a review article for that reason, but also due to the long history section at the beginning. Thus, I think more of the research on DBS of these areas needs to be covered and more in line of what a review article would have as opposed to a textbook type of writing style.

Thank you very much for the comments and suggestions. The objective of our work is to describe the anatomical structures and their connections that are related to the process of emotions and behavior. The understanding of these structures is vital to understand the choice of the different therapeutic targets for DBS. Description of outcomes and adverse effects in DBS, are beyond our scope and we believe they would deserve a full new (and different of this) article.

On the other hand, the historical context seems to us to be important, since it emphasizes the advances in the study of emotions. Students may benefit from this type of information

Minor comments:

Overall, the article was well written. However, there are some instances of various types of typos, wording, and writing style as well as formatting that should be changed.

  1. In a large number of places in the paper some words are in a different font than the others. Here are some examples that appear to be a different font, although there are likely more that I missed. Please correct these and any other instances of this.

Done

  1. Line 30 “In contrast”

Done

  1. Line 64 “Gall’s”

Done

  1. Line 102 “behavior”

Done

  1. Line 117 “has been”

Done

  1. Line 234 “form”

Done

  1. There are many instances in the paper where the authors have a paragraph that is one sentence. There are a few others with 2 sentences. A paragraph in general shouldn’t be one sentence. These can be combined into another paragraph and would logically fit in one of the surrounding paragraphs. Considering changing these. Examples include paragraphs associated with lines 39-41, 117-119, 152-154, 277-278, 290-292, 363-364, 366-367, 387-389, 391-393, and 453-455.

The reviewer is right. We have added a sentence to clarify this important point at Figure 1 legend.

  1. A few times the manuscript is referred to as “this study”. I think referring to it as “this review” would be better. Examples are lines 39 and 435.

Changed

  1. Bibliography: spacing in line 536 is off. Furthermore, some articles have all caps for titles of studies (e.g. referencse 42, 57, 58, 59, 64) whereas most others do not. Check all bibliography formatting.

Done

  1. Lines 68-76. Why not call him Phineus Gage by name since this is such a historic example. Students may not have heard of this and want to find out about the story. Mentioning his name would be helpful.

Changed

  1. Line 140 should “striate” read “striatal”??

You are right, the expression is: The cortico-striate-thalamo-cortical circuit

  1. Figure 1. Why does the black line from the SNc change to orange? Why not just be orange the whole way. Better yet since this pathway has two branches and is excitatory to the direct pathway and inhibitory to the indirect pathway of the striatum (due to different post synaptic receptors) why not have 2 separate lines the whole way from the SNc??? In addition, wouldn’t it be better to point out the direct, indirect, and hyperdirect pathways on this figure? They are depicted but not labeled. Based on the importance of knowing the physiology that this article discusses it seems like that should be on there.

The reviewer is right. We have added a sentence clarifying this point because at Figure 1 legend.

  1. Figure 3. Would “afferents” or “afferent pathways” be better to put on the figure than “afferences”? Analogous comment for efferences.

Changed

  1. If possible figures 2, 3, and 4 could be bigger.

Of course, but this is more a matter of edition.

Reviewer 3 Report

Thank you for giving me the opportunity to review this manuscript. I think this manuscript is interesting and organized, but it is still necessary to revise the manuscript.   1) Please refer to previous studies more and emphasize the novelty and significance of this study.   a) Miguel A. Faria, Jr. Violence, mental illness, and the brain – A brief history of psychosurgery: Part 1 – From trephination to lobotomy. Surgical Neurology International 2013, 4:49.   b) Miguel A. Faria, Jr. Violence, mental illness, and the brain – A brief history of psychosurgery: Part 2 – From the limbic system and cingulotomy to deep brain stimulation. Surgical Neurology International 2013, 4:75   2) The authors reviewed OCD, refractory depression, erethism, anorexia nervosa, and drug addition. However, rTMS for refractory depression has been used worldwide, and FDA approved deep rTMS for clinical use in patients with OCD or nicotine dependence. Please review these other neuromodulation in clinical practice and why DBS in patients with OCD or drug addiction are warranted. Furthermore, erethism is not included in mental disorder, I think. Please refer to what kind of erethism DBS can target for.   3)  I think both OCD, refractory depression, erethism, anorexia nervosa, and drug addition are quite heterogeneous. What kind of patients with OCD, refractory depression, erethism, anorexia nervosa, and drug addition are eligible for DBS? What kind of biomarkers are promising in treating these disorders  by DBS? What aspects of OCD, refractory depression, erethism, anorexia nervosa, and drug addition  are promisin as novel targets?   I think it is better to revise the manuscript.

Author Response

Dear reviewer:

Thank you for the comments concerning our manuscript. Those comments are all valuable and very helpful for revising and improving our paper. We have studied comments carefully and have made correction which we hope meet with approval. We will reply one by one as follows and the part that replies to you is marked in red of all revisions in the revised manuscript.

Thank you for giving me the opportunity to review this manuscript. I think this manuscript is interesting and organized, but it is still necessary to revise the manuscript.   1) Please refer to previous studies more and emphasize the novelty and significance of this study.   a) Miguel A. Faria, Jr. Violence, mental illness, and the brain – A brief history of psychosurgery: Part 1 – From trephination to lobotomy. Surgical Neurology International 2013, 4:49.   b) Miguel A. Faria, Jr. Violence, mental illness, and the brain – A brief history of psychosurgery: Part 2 – From the limbic system and cingulotomy to deep brain stimulation. Surgical Neurology International 2013, 4:75  

According to the Brain Sciences Guidelines for Review Reports (MDPI | Guidelines for Reviewers): “Reviewers must not recommend excessive citation of their work (self-citations), another author’s work (honorary citations) or articles from the journal where the manuscript was submitted as a means of increasing the citations of the reviewer/authors/journal. You can provide references as needed, but they must clearly improve the quality of the manuscript under review”. We are not completely sure that these suggestions (from the same author) did not violate this Editorial’s recommendation.

2) The authors reviewed OCD, refractory depression, erethism, anorexia nervosa, and drug addition. However, rTMS for refractory depression has been used worldwide, and FDA approved deep rTMS for clinical use in patients with OCD or nicotine dependence. Please review these other neuromodulation in clinical practice and why DBS in patients with OCD or drug addiction are warranted.

As stated at Introduction “In this review, we summarize the main neural networks and anatomical structures implicated in affective disorders and the rational to target them for deep brain stimulation surgery (DBS).” Although we acknowledge the useful and relevance of rTMS, it’s clearly out of our interest, therefore, we are do not need to address issues or techniques not claimed as our goal.

Furthermore, erethism is not included in mental disorder, I think. Please refer to what kind of erethism DBS can target for.  

The reviewer is right and we have modified accordingly to clarify this question. We have changed erethism by aggressivity.

3)  I think both OCD, refractory depression, erethism, anorexia nervosa, and drug addition are quite heterogeneous. What kind of patients with OCD, refractory depression, erethism, anorexia nervosa, and drug addition are eligible for DBS? What kind of biomarkers are promising in treating these disorders  by DBS? What aspects of OCD, refractory depression, erethism, anorexia nervosa, and drug addition  are promisin as novel targets?   I think it is better to revise the manuscript.

These are extremely interesting questions but none of them are of our interest for this review. Therefore, we think that next works can address these interesting topics.

Round 2

Reviewer 1 Report

The authors did not accept most of the suggestions given by the Reviewer. 

Author Response

Dear reviewer:

We tried to address your comments with the highest integrity and honesty possible. Discrepancy is not only frequent but a good practice in Science. We tried to explain and defense our position with rational arguments.

Thank you very much for your help to improve the manuscript.

Reviewer 2 Report

The authors seem to have made all the changes I recommended. I think the paper is now publishable.

Some minor English editing will probably be required.

Author Response

Dear reviewer:

Thank you very much for your comments and for your help to improve the manuscript.

Reviewer 3 Report

I think this manuscript is still not ethical, scientific or clinically meaningful.

Author Response

Dear reviewer:

We tried to address your comments with the highest integrity and honesty possible. Discrepancy is not only frequent but a good practice in Science. We tried to explain and defense our position with rational arguments.

We have added a paragraph at Discussion about the use of rTMS in psychiatric illness.

We can understand that you do not like the manuscript and even you find it scarcely useful for readers, but, let us comment your sentence. The Merriam-Webster’s Dictionary defines the adjective meaningful as having a meaning or purpose. We could accept that maybe our work would be badly wrote, confusedly described and even poorly relevant for the field (although this point finally depends on the readers) but, evidently has a clinical and scientific purpose “we summarize the main neural networks and anatomical structures implicated in affective disorders and the rational to target them for deep brain stimulation surgery (DBS)” and in no case can be defined as meaningless. Regarding the expression “absence of ethical meaning” applied to a review scientific paper when experiments are not described and the primary material are other articles, is difficult to understand and impossible to accept.